




4     **Hydraulic and transport parameter assessment using column infiltration experiments**

6         A. Younes[1,2,3], T.A. Mara[4], M. Fahs[1], O. Grunberger[2], Ph. Ackerer[*,1]

7         [1] LHyGES, Université de Strasbourg/EOST, CNRS, 1 rue Blessig, 67084 Strasbourg, France.
8         [2] IRD UMR LISAH, F-92761 Montpellier, France.
9         [3] LMHE, Ecole Nationale d'Ingénieurs de Tunis, Tunisie
10        [3] Université de La Réunion, PIMENT, 15 Avenue René Cassin, BP 7151, 97715 Moufia, La Réunion.

Submitted to
* Contact person: Ph. Ackerer
E-mail: ackerer@unistra.fr





*Abstract*
In the present work, we study the quality of the statistical calibration of hydraulic and
transport soil properties using an infiltration experiment in which, over a given period, tracer-
contaminated water is injected into a laboratory column filled with a homogeneous soil. The
numerical model is based on the Richards' equation for solving water flow and the advection-
dispersion equation for solving solute transport. Several state variables (e.g., water content,
solute concentration, pressure head) are measured during the experiment. Statistical
calibration of the computer model is then carried out for different data sets and injection
scenarios with the DREAM$_{(ZS)}$ Markov Chain Monte Carlo sampler. The results show that the
injection period has a significant effect on the quality of the estimation, in particular, the
posterior uncertainty range. The hydraulic and transport parameters of the investigated soil
can be estimated from the infiltration experiment using the concentration and cumulative
outflow, which are measured non-intrusively. A significant improvement of the identifiability
of the parameters is observed when the pressure data from measurements taken inside the
column are also considered in the inversion.

**Keywords**
Infiltration experiment, Richards' equation, Statistical calibration, Markov Chain Monte
Carlo, Uncertainty ranges.





## 1. Introduction

The soil parameters that influence water flow and contaminant transport in unsaturated zones are not generally known *a priori* and have to be estimated by fitting model responses to observed data. Several studies have demonstrated that unsaturated soil hydraulic parameters can be (more or less accurately) estimated from dynamic flow experiments (*e.g*., Hopmans et al., 2002; Vrugt et al., 2003a; Durner and Iden, 2011; Younes et al., 2013). Inoue et al. (2000) showed that both hydraulic and transport parameters can be assessed by the combination of flow and transport experiments. Indeed, the simultaneous estimation of hydraulic and transport properties yields smaller estimation errors for model parameters than the sequential inversion of hydraulic properties from the water content and/or pressure head followed by the inversion of transport properties from concentration data (Misra and Parker, 1989).

In the present work, we consider the flow and the transport of an inert solute injected into a laboratory column filled with a homogeneous sandy clay loam soil. The flow-transport model is described by the Richards' equation (RE) for water flow and the advection dispersion equation for solute transport. The Mualem-van Genuchten (MvG) models (Mualem 1976, van Genuchten 1980) are chosen to describe the retention curve and to relate the hydraulic conductivity of the unsaturated soil to the water content. The estimation of hydraulic and transport parameters is performed in a Bayesian framework using the Markov Chain Monte Carlo (MCMC) sampler (Vrugt and Bouten, 2002; Vrugt et al., 2008) for two injection periods and different data measurement scenarios. Unlike classical parameter optimization algorithms, the MCMC approach provides parameter joint probability distributions, which are useful for the quality assessment of the estimation. Indeed, MCMC samples can be used to summarize parameter uncertainties and to perform predictive uncertainty (Ades and Lu, 2003).





Soil parameters are usually investigated using multistep outflow experiments (e.g., Eching
and Hopmans, 1993; Eching et al., 1994; van Dam et al., 1994) or continuously changing
time-varying boundary conditions (Durner et al., 1999). Multistep outflow experiments are
among the most popular laboratory methods (Hopmans et al., 2002). However, their
application is limited by expensive measurement equipment (Nasta et al., 2011).
In this work, hydraulic soil parameters are investigated using an infiltration experiment in a
1.2 m long laboratory column, which is the standard scale for these types of experiments. The
column, which is initially hydrostatic and free of solute, is filled with a homogeneous sandy
clay loam soil. Continuous flow and solute injection are performed during a time period $T_{inj}$ at
the top of the column and with a zero pressure head at the bottom. The unknown parameters
for the water flow are $k_s$ $[LT^{-1}]$, the saturated hydraulic conductivity; $\theta_s$ $[L^3L^{-3}]$, the
saturated water content; $\theta_r$ $[L^3L^{-3}]$, the residual water content; and $\alpha$ $[L^{-1}]$ and $n$ $[-]$, the
MvG shape parameters. The only unknown parameter of the tracer transport is the
longitudinal dispersivity, $a_L$ $[L]$.
Several scenarios corresponding to different sets of measurements are investigated to address
the following questions:
1) Can we obtain an appropriate estimation of all flow and transport parameters from the
tracer-infiltration experiment, even though only moderately dry conditions are used?
2) What is the optimal set of measurements for the estimation of all the parameters? Can
we use only non-intrusive measurements (cumulative outflow and concentration
breakthrough curve) or are intrusive measurements required, such as the analysis of
the pressure head and/or water content inside the column?
3) Does the duration of the injection period $T_{inj}$ have an impact on the identification of
the parameters?





Synthetic scenarios are considered in the sequel in which data from numerical simulations are
manipulated to avoid the uncontrolled noise of experiments that could bias the conclusions.
The paper is organized as follows. The mathematical models describing flow and transport in
the unsaturated zone are detailed in section 2. Section 3 describes the MCMC Bayesian
parameter estimation procedure used in the DREAM$_{(ZS)}$ sampler. Section 4 presents the
different investigated scenarios and discusses the results of the calibration in terms of mean
parameter values and uncertainty ranges for each scenario. Conclusions are given in section 5.

## 2. Unsaturated flow-transport model

We consider a uniform soil profile in the column and an injection of a solute tracer such as
bromide, as described in Mertens et al. (2009). The unsaturated water flow in the vertical soil
column is modeled with the one-dimensional pressure head form of the RE:
$$\begin{cases} \left( c(h) + S_s \dfrac{\theta}{\theta_s} \right) \dfrac{\partial h}{\partial t} = \dfrac{\partial q}{\partial z}, \\ q = K(h)\left( \dfrac{\partial h}{\partial z} - 1 \right) \end{cases} \quad (1)$$

where $h$ [L] is the pressure head; $q$ [LT$^{-1}$] is the Darcy velocity; $z$ [L] is the depth, measured
as positive in the downward direction; $S_s$ (-) is the specific storage; $\theta$ and $\theta_s$ [L$^3$.L$^{-3}$] are the
actual and saturated water contents, respectively; $c(h)$ [L$^{-1}$] is the specific moisture capacity;
and $K(h)$ [L T$^{-1}$] is the hydraulic conductivity. The latter two parameters are both functions
of the pressure head. In this study, the relations between the pressure head, conductivity and
water content are described by the following standard models of Mualem (1972) and van
Genuchten (1980):






$$S_e(h) = \frac{\theta(h) - \theta_r}{\theta_s - \theta_r} = \begin{cases} \dfrac{1}{\left(1 + |\alpha h|^n\right)^m} & h < 0 \\ 1 & h \geq 0 \end{cases}$$ (2)

$$K(S_e) = K_s S_e^{1/2} \left[1 - \left(1 - S_e^{1/m}\right)^m\right]^2 ,$$

where $S_e$ (-) is the effective saturation, $\theta_r$ [L$^3$ L$^{-3}$] is the residual water content, $K_s$ [L T$^{-1}$] is
the saturated hydraulic conductivity, and $m = 1 - 1/n$, $\alpha$ [L$^{-1}$] and $n$ (-) are the MvG shape
parameters.
The tracer transport is governed by the following convection-dispersion equation:

$$\frac{\partial(\theta C)}{\partial t} + \frac{\partial(qC)}{\partial z} - \frac{\partial}{\partial z}\left(\theta D \frac{\partial C}{\partial z}\right) = 0 ,$$ (3)

where $C$ [ML$^{-3}$] is the concentration of the tracer, $D$ [L$^2$ T$^{-1}$] is the dispersion coefficient in
which $D = a_l q + d_m$ and $a_l$ [L] is the dispersivity coefficient of the soil and $d_m$ [L$^2$ T$^{-1}$] is
the molecular diffusion coefficient, which is set as 1.04 10$^{-4}$ cm$^2$/min.
The initial conditions are as follows: a hydrostatic pressure distribution with zero pressure
head at the bottom of the column $(z = L)$ and a solute concentration of zero inside the whole
column. An infiltration with a flux $q_{inj}$ of contaminated water with a concentration $C_{inj}$ is
then applied at the upper boundary condition (z = 0) during a period $T_{inj}$. Hence, the boundary
conditions at the top of the column can be expressed as:
$for\ 0 < t \leq T_{inj}$ $\begin{cases} K\left(\dfrac{\partial h}{\partial z} - 1\right) = q_{inj} \\ \theta D \dfrac{\partial C}{\partial z} + qC = q_{inj} C_{inj} \end{cases}$ $for\ t > T_{inj}$ $\begin{cases} K\left(\dfrac{\partial h}{\partial z} - 1\right) = 0 \\ C_{inj} = 0 \end{cases}$ (4)
A zero pressure head is maintained at the lower boundary $(z = L)$ of the column and a zero
concentration gradient is used as the lower boundary condition for the solute transport.





$$(h)_{z=l} = 0 \qquad \left(\frac{\partial C}{\partial z}\right)_{z=l} = 0 \qquad\qquad (5)$$
In the sequel, the infiltration rate and the injected solute concentration are $q_{inj} = 0.015$ cm/min
and $C_{inj} = 1$ g/cm$^3$, respectively. The system (1)-(3) is solved using the finite volume method
for both flow and transport spatial discretization. A uniform mesh of 600 cells is employed.
Temporal discretization is performed with the high-order method of lines (MOL) (e.g., Miller
et al., 1998; Tocci et al., 1997; Fahs et al., 2009). Error checking, robustness, order selection
and adaptive time step features, available in sophisticated solvers, are applied to the time
integration of partial differential equations in the MOL (Tocci et al., 1997). The MOL has
been successfully used to solve RE in many studies (e.g., Farthing et al., 2003; Miller et al.,
2006; Li et al., 2007; Fahs et al., 2009).
The unknown parameters for the water flow are $k_s$, $\theta_s$, $\theta_r$ and the MvG shape parameters $\alpha$
and $n$. The only unknown parameter of the tracer transport is the longitudinal dispersivity $a_L$
. Hence, the total vector of parameters is $\xi = (k_s, \theta_s, \theta_r, \alpha, n, a_L)$. A reference solution is
generated using the following parameter values (corresponding to a sandy clay loam soil):
$k_s = 50\, cm/day$, $\theta_s = 0.43$, $\theta_r = 0.09$, $\alpha = 0.04\, cm^{-1}$, $n = 1.4$ and $a_l = 0.2\, cm$. Four types of
observations are deduced from the results of the simulation, which include the following: the
pressure head and water content near the surface (5 cm below the top of the column) as well
as the cumulative outflow and the breakthrough concentration at the output of the column.
The vector of observations $y_{mes}$ is formed by the four data series, which are independently
corrupted with a normally distributed noise using the following standard deviations: $\sigma_h = 1\, cm$
for the pressure head, $\sigma_\theta = 0.02$ for the water content, $\sigma_Q = 0.1$cm for the cumulative
outflow and $\sigma_C = 0.01$ g/cm$^3$ for the exit concentration.





**3. Bayesian parameter estimation**

The flow-transport model is used to analyze the effects of different measurement sets on parameter identification. For this purpose, we adopt a Bayesian approach that involves the parameter joint posterior distribution (Vrugt et al., 2008). The latter is assessed with the DREAM$_{(ZS)}$ MCMC sampler (Laloy and Vrugt, 2012). This software generates random sequences of parameter sets that asymptotically converge toward the target joint posterior distribution (Gelman et al., 1997). Thus, if the number of runs is sufficiently high, the generated samples can be used to estimate the statistical measures of the posterior distribution, such as the mean and variance among other measures.

The Bayes theorem states that the probability density function of the model parameters conditioned onto data can be expressed as:

$$p\left(\xi \mid \boldsymbol{y}_{mes}\right) \propto p\left(\boldsymbol{y}_{mes} \mid \xi\right) p\left(\xi\right) \tag{6}$$

where $p\left(\xi \mid \boldsymbol{y}_{mes}\right)$ is the likelihood function measuring how well the model fits the observations $\boldsymbol{y}_{mes}$, and $p\left(\xi\right)$ is the prior assumption of the parameter before the observations are made. In this work, a Gaussian distribution defines the likelihood function because the observations are simulated and corrupted with Gaussian errors. In addition, independent uniform priors are considered. Hence, the parameter posterior distribution is expressed as:

$$p\left(\xi / \boldsymbol{y}_{mes}\right) \propto exp\left(-\frac{SS_h\left(\xi\right)}{2\sigma_h^2} - \frac{SS_\theta\left(\xi\right)}{2\sigma_\theta^2} - \frac{SS_Q\left(\xi\right)}{2\sigma_Q^2} - \frac{SS_C\left(\xi\right)}{2\sigma_C^2}\right) \tag{7}$$

where $SS_h\left(\xi\right)$, $SS_\theta\left(\xi\right)$, $SS_Q\left(\xi\right)$ and $SS_C\left(\xi\right)$ are the sums of the squared differences between the observed and modeled data of the pressure head, water content, cumulative outflow and output concentration, respectively. For instance, $SS_h\left(\xi\right) = \sum_{k=1}^{Nh}\left(h_{mes}^{(k)} - h_{mod}^{(k)}\left(\xi\right)\right)^2$,




which includes the observed and predicted pressure heads $h_{mes}^{(k)}$ and $h_{mod}^{(k)}$ at time $t_k$ and the
number of pressure head observations $Nh$.
Bayesian parameter estimation is performed hereafter with the DREAM$_{(ZS)}$ software (Laloy
and Vrugt, 2012), which is an efficient MCMC sampler. DREAM$_{(ZS)}$ computes multiple sub-
chains in parallel to thoroughly explore the parameter space. Archives of the states of the sub-
chains are also stored and used to allow a strong reduction of the "burn-in" period in which
the sampler generates individuals with poor performances. Taking the last 25% of individuals
of the MCMC (when the chains have converged) yields multiple sets of parameters, $\xi$, that
adequately fit the model onto observations. These sets are then used to estimate the updated
parameter distributions, the pairwise parameter correlations and the uncertainty of the model
predictions. As suggested in Vrugt et al. (2003b), the posterior distribution becomes
stationary if the Gelman and Ruban (1992) criterion is $\leq 1.2$.
**4. Results and discussion**
In this section, the identifiability of the parameters is investigated for different scenarios of
measurement sets and for two periods of injections. In all cases, the MCMC sampler was run
with 3 simultaneous chains for a total number of 50000 runs. Depending on the scenario, the
MCMC required between 5000 and 20000 model runs to reach convergence. The last 25% of
the runs that adequately fit the model onto observations are used to estimate the updated
probability density function (pdf).

*4.1. Reference solution and data measurements*
The reference solutions obtained from solving the flow-transport problems (1)-(3) using the
parameters given above are shown in Fig. 1 to 6. The pressure head at 5 cm, at the top of the
column (Fig. 1), increases quickly from its initial hydrostatic negative value (approximately -





115 cm) and reaches a plateau (-1.75 cm) during the injection period. After the injection is
finished, it progressively decreases due to the drainage caused by the gravity effect. A similar
behavior is observed for the water content at the same location (Fig. 2), where the value of the
plateau is close to the saturation value. The cumulative outflow (Fig. 3) starts to increase at
approximately 1000 min after the beginning of the injection. It shows an almost linear
behavior until 5500 min. It then slowly increases with an asymptotic behavior due to the
natural drainage after the end of the injection. Fig. 4 displays the water saturation as a
function of the pressure head. It is worth noting that only a few parts of this curve are
described during the infiltration experiment. Indeed, only moderate dry conditions are
established because the minimum pressure head reached in the column is -120 cm, which
corresponds to the initial pressure head near the top of the column.
The breakthrough concentration curve (Fig. 5) shows a sharp front, which starts shortly after
3000 min. If the injection of both water and contaminant are stopped once the solute reaches
the output, i.e., after an injection period of 3000 min, the breakthrough curve exhibits a
smoother progression (Fig. 6).
The observed data, which are used as conditioning information for model calibration, are also
shown in Fig. 1to 6. Fig. 2 shows that the water content is more affected by the perturbation
of data than by the pressure head and cumulative outflow because (*i*) we mimic the relative
importance of the measurement errors of the water content due to time-domain-reflectometry
probes and (*ii*) the weak variation of the water content during the infiltration experiment. The
perturbation of the breakthrough curve is relatively small because output concentrations can
be accurately measured. The perturbations of the pressure head and cumulative outflow seem
weak because of the large variation of these variables during the experiment.





### 4.2. Results of the parameter estimation

The uncertainty model parameters are assumed to be distributed uniformly over the ranges reported in Table 1. This table also lists the reference values used to generate data observations before perturbation. Seven scenarios, corresponding to different sets of measurements for the estimation of the soil parameters, are considered (Table 2).

The MCMC results of the seven studied scenarios are given in Figs. 8 to 13. The "on-diagonal" plots in these figures display the inferred parameter distributions, whereas the "off-diagonal" plots represent the pairwise correlations in the MCMC sample. If the drawings are independent, non-sloping scatterplots should be observed. However, if a good value of a given parameter is conditioned by the value of another parameter, then their pairwise scatterplot should show a narrow sloping stripe. To facilitate the comparison between the different scenarios, Fig. 14 to 19 show the mean and the 95% confidence intervals of the final MCMC sample that adequately fit the model onto observations for each scenario, and Table 3 summarizes the pairwise parameter correlations.

Fig. 7 shows the inferred distributions of the parameters identified with the MCMC sampler using only the pressure and cumulative outflow measurements (scenario 1). The parameters $k_s$, $\alpha$ and $n$ are well estimated; their prior intervals of variation are strongly narrowed and they essentially show bell-shaped posterior distributions. Parameter $k_s$ is strongly correlated to $\alpha$ (0.94) and $n$ (-0.97). Because the water retention relationship depends on the difference between $\theta_s$ and $\theta_r$, these parameters are strongly correlated (0.96) and cannot be identified. The dispersivity coefficient $a_l$ has not been identified.

The MCMC results (Fig. 8) show that $\theta_r$ strongly correlates to $k_s$ (-0.94) and $n$ (0.98) when water content measurements are added into the model (scenario 2). The parameter $k_s$ remains strongly related to $\alpha$ (0.94) and $n$ (-0.98). Although the water content data are subject to





relatively high measurement errors, a good estimation is obtained for $\theta_s$ and $\theta_r$. The
parameters $k_s$, $\alpha$ and $n$ are estimated with the same accuracy as for the first scenario.
When the concentration measurements are also considered (scenario 3), the results depicted in
Fig. 9 show very significant correlations between $k_s$ and $\theta_r$ (-0.94), $k_s$ and $\alpha$ (0.91), $k_s$ and
$n$ (-0.97) and $n$ and $\theta_r$ (0.99). The posterior uncertainty ranges of $k_s$, $\alpha$, $n$ and $\theta_r$ are
similar to the previous scenarios. Those of $\theta_s$ and $a_l$ are strongly reduced, leading to a good
identification of these parameters when using $C$ measurements (Fig. 15 and 19). A better
estimate of the saturated water content is expected because advective transport is a function of
this variable.
The measurements of the water content are not considered in the inversion procedure of
scenario 4. This scenario leads to the same quality of the estimation for the parameters $k_s$, $\theta_r$,
$\alpha$ and $n$ (Fig. 14, 16, 17, 18) and similar correlations between the parameters as in the
previous scenario. This result shows that the intrusive water content measurements, which are
subject to more measurement errors than the output concentration, are not required if the
output concentration is measured. Compared with the results of scenario 2, it can be
concluded that better parameter estimations are obtained using $h$, $Q$ and $C$ data than using
$h$, $Q$ and $\theta$ data, especially for $\theta_s$. Therefore, using $C$ instead of $\theta$ measurements in
combination with $h$ and $Q$ measurements allows the estimation of $a_l$ and leads to a better
estimate of $\theta_s$.
The pressure head, cumulative outflow and concentration measurements are used in the
estimation procedure of scenario 5, but the injection period is now reduced to $T_{inj} = 3000\,\text{min}$.
The obtained results (Fig. 11) show the same correlations between the parameters as for
$T_{inj} = 5000\,\text{min}$. For the parameters $k_s$, $\theta_s$, $\theta_r$, $\alpha$ and $n$, almost the same mean estimates are





obtained as for scenario 4. However, the parameters are better identified (Fig. 14 to 18).
Indeed, the uncertainty of these parameters is smaller because the credible interval is reduced
by a factor of 25% for $k_s$, 8% for $\theta_s$, 26% for $\theta_r$, 10% for $\alpha$ and 25% for $n$ when compared
to the results obtained for $T_{inj} = 5000\,\text{min}$. The parameter $a_l$ is also estimated much better
than in the previous scenario. Its mean value approaches the reference solution and the
posterior uncertainty range is reduced by approximately 75% (Fig. 19).
The pressure head measurements are removed in scenario 6 and only non-intrusive
measurements ($Q$ and $C$ data) are used with an injection period of $T_{inj} = 5000\,\text{min}$. The
results depicted in Fig. 12 show high correlations only between $k_s$ and $n$ (-0.95) and $\theta_r$ and
$n$ (0.95). Compared with the results of scenario 4, which also considers the pressure data, $k_s$
is poorly estimated (the mean value is less close to the reference value and the credible
interval is 27% larger). The mean estimated values for $\theta_r$ and $n$ also degraded (less close to
the reference solution), although their confidence intervals are similar to those of scenario 4
(Fig. 16, 18). The estimated mean value of parameter $\alpha$ is similar to that in scenario 4.
However, its uncertainty is much larger because the credible interval is 77% larger (Fig. 19).
The parameters $\theta_s$ and $a_l$ are estimated as well in scenario 4 (in terms of mean estimated
value and credible interval).
The last scenario (scenario 7) is similar to the previous one, but the injection period is reduced
to $T_{inj} = 3000\,\text{min}$. The results depicted in Fig. 13 show similar correlations between the
parameters as for $T_{inj} = 5000\,\text{min}$. However, a significant improvement is observed for the
mean estimated values, which approach the reference solution for $k_s$, $\theta_r$, $n$ and $a_l$ (Fig. 14,
16, 18, 19). The uncertainties of $k_s$, $\alpha$ and $a_l$ are also reduced by approximately 40%, 15%
and 70%, respectively. The parameter $\theta_s$ is estimated as well in scenario 6.



**5. Conclusions**
In this work, hydraulic and transport soil parameters have been estimated using an infiltration
experiment performed in a laboratory column filled with sandy clay loam soil, which was
subjected to continuous flow and solute injection over a period $T_{inj}$. Parameter estimation was
performed for different scenarios of data measurements in a Bayesian framework using the
DREAM$_{(zs)}$ MCMC sampler (Laloy and Vrugt, 2012).

The results reveal the following conclusions:
1. All hydraulic and transport parameters can be appropriately estimated from the

described infiltration experiment. However, the accuracy differs and depends on the

type of measurement and the duration of the injection $T_{inj}$, even if the water content

remains close to saturated conditions.

2. The use of concentration measurements at the column outflow, in addition to

traditional measured variables (water content, pressure head and cumulative outflow),

reduces the correlation between the hydraulic parameters and their uncertainties,

especially that of the saturated water content.

3. The saturated hydraulic conductivity is estimated with the same order of accuracy,

independent of the observed variables.

4. The estimation of the dispersivity is sensitive to the injection duration.
5. A better identifiability of the soil parameters is obtained using $C$ instead of $\theta$

measurements, in combination with $h$ and $Q$ data.

6. Using only non-intrusive measurements (cumulative outflow and output

concentration) allows the satisfactory estimation of all parameters. The uncertainty of

the parameters significantly decreases when the injection of water and solute is

maintained for a limited period.






317 This last point has practical applications for designing simple experimental setups dedicated

318 to the estimation of hydrodynamic and transport parameters for unsaturated flow in soils. The

319 setup has to be appropriately equipped to measure the cumulative water outflow (e.g.,

320 weighing machine) and the solute breakthrough at the column outflow (e.g., flow through

321 electrical conductivity). The injection should be stopped as soon as the solute concentration

322 reaches the outflow. The accuracy of the estimation of $\theta_r$, $\alpha$ and $n$ can be improved by

323 adding pressure measurements inside the column, close to the injection.


325 **Acknowledgments**

326 The authors are grateful to the French National Research Agency, which funded this work

327 through the program AAP Blanc - SIMI 6 project RESAIN (n° ANR-12-BS06-0010-02).




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






**List of table captions**

Table 1. Prior lower and upper bounds of the uncertainty parameters and reference values.

Table 2. Measurement sets and injection periods for the different scenarios. The pressure head
$h$ and the water content $\theta$ are measured at 5 cm from the top of the column. The cumulative
outflow $Q$ and the concentration $C$ are measured at the exit of the column.

Table 3. Summary of the pairwise parameter correlations.






| Parameters | Lower bounds | Upper bounds | Reference values |
|---|---|---|---|
| $k_s$ [cm min$^{-1}$] | 0.025 | 0.1 | 0.0347 |
| $\theta_s$ [-] | 0.3 | 0.5 | 0.43 |
| $\theta_r$ [-] | 0.05 | 0.2 | 0.09 |
| $\alpha$ [cm$^{-1}$] | 0.01 | 0.3 | 0.04 |
| $n$ [-] | 1.2 | 5 | 1.4 |
| $a_l$ [cm] | 0.05 | 0.6 | 0.2 |


Table 1. Prior lower and upper bounds of the uncertainty parameters and reference values.




| Scenario | Measured variables | | | | injection period | |
|---|---|---|---|---|---|---|
| | $h$ | $\theta$ | $Q$ | $C$ | $T_{inj} = 5000\,\text{min}$ | $T_{inj} = 3000\,\text{min}$ |
| 1 | ν | | ν | | ν | |
| 2 | ν | ν | ν | | ν | |
| 3 | ν | ν | ν | ν | ν | |
| 4 | ν | | ν | ν | ν | |
| 5 | ν | | ν | ν | | ν |
| 6 | | | ν | ν | ν | |
| 7 | | | ν | ν | | ν |


Table 2. Measurement sets and injection periods for the different scenarios. The pressure head
$h$ and the water content $\theta$ are measured at 5 cm from the top of the column. The cumulative
outflow $Q$ and the concentration $C$ are measured at the exit of the column.










| Scenario | | | | | |
|---|---|---|---|---|---|
| 1 | $(k_s, n)$ | $(k_s, \alpha)$ | | | $(\theta_r, \theta_s)$ |
| 2 | $(k_s, n)$ | $(k_s, \alpha)$ | $(k_s, \theta_r)$ | $(\theta_r, n)$ | |
| 3 | $(k_s, n)$ | $(k_s, \alpha)$ | $(k_s, \theta_r)$ | $(\theta_r, n)$ | |
| 4 | $(k_s, n)$ | $(k_s, \alpha)$ | $(k_s, \theta_r)$ | $(\theta_r, n)$ | |
| 5 | $(k_s, n)$ | $(k_s, \alpha)$ | $(k_s, \theta_r)$ | $(\theta_r, n)$ | |
| 6 | $(k_s, n)$ | | | $(\theta_r, n)$ | |
| 7 | $(k_s, n)$ | | | $(\theta_r, n)$ | |

Table 3. Summary of the pairwise parameter correlations.




**List of figure captions**

Fig. 1. Reference pressure head at 5 cm from the soil surface. Solid lines represent model outputs and dots represent the sets of perturbed data serving as conditioning information for model calibration.

Fig. 2. Reference water content at 5 cm from the soil surface [see Fig. 1 caption ].

Fig. 3. Reference cumulative outflow [see Fig. 1 caption ].

Fig. 4. Reference retention curve for the infiltration experiment [see Fig. 1 caption ].

Fig. 5. Reference breakthrough output concentration for $T_{inj} = 5000$. [see Fig. 1 caption ].

Fig. 6. Reference breakthrough output concentration for $T_{inj} = 3000$ min. [see Fig. 1 caption ].

Fig. 7. MCMC solutions for the transport scenario 1. The diagonal plots represent the inferred posterior probability distribution of the model parameters. The off-diagonal scatterplots represent the pairwise correlations in the MCMC drawing.

Fig. 8. MCMC solutions for transport scenario 2 [see Fig. 7 caption ].

Fig. 9. MCMC solutions for transport scenario 3 [see Fig. 7 caption ].

Fig. 10. MCMC solutions for transport scenario 4 [see Fig. 7 caption ].

Fig. 11. MCMC solutions for transport scenario 5 [see Fig. 7 caption ].

Fig. 12. MCMC solutions for transport scenario 6 [see Fig. 7 caption ].

Fig. 13. MCMC solutions for transport scenario 7 [see Fig. 7 caption ].

Fig. 14. Posterior mean values and 95% confidence intervals of the saturated hydraulic conductivity for the different scenarios.

Fig. 15. Posterior mean values and 95% confidence intervals of the saturated water content for the different scenarios.

Fig. 16. Posterior mean values and 95% confidence intervals of the residual water content for the different scenarios.

Fig. 17. Posterior mean values and 95% confidence intervals of the shape parameter □ for the different scenarios.

Fig. 18. Posterior mean values and 95% confidence intervals of the shape parameter n for the different scenarios.

Fig. 19. Posterior mean values and 95% confidence intervals of dispersivity for the different scenarios.






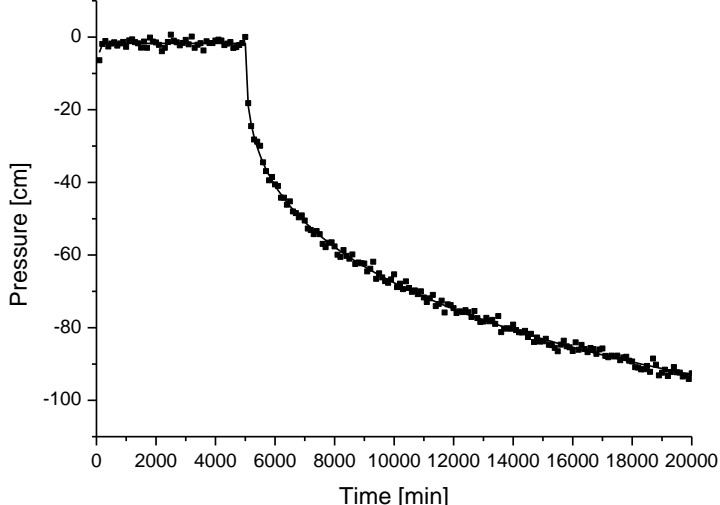

Fig. 1. Reference pressure head at 5 cm from the soil surface. Solid lines represent model
outputs and dots represent the sets of perturbed data serving as conditioning information for
model calibration.

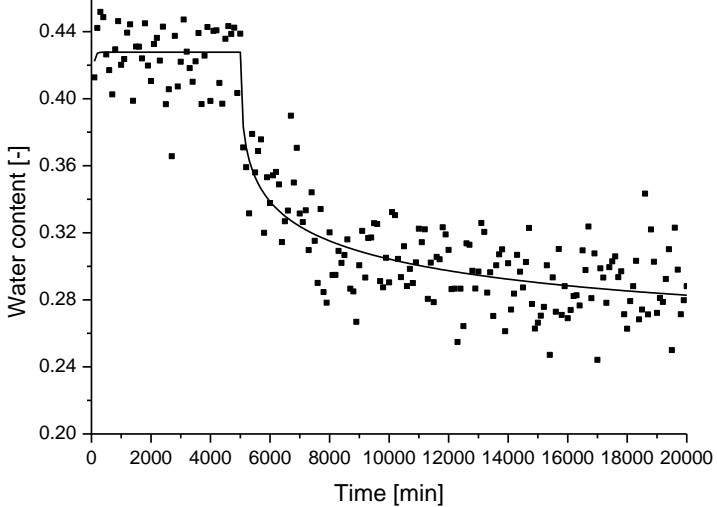

Fig. 2. Reference water content at 5 cm from the soil surface [see Fig. 1 caption ].





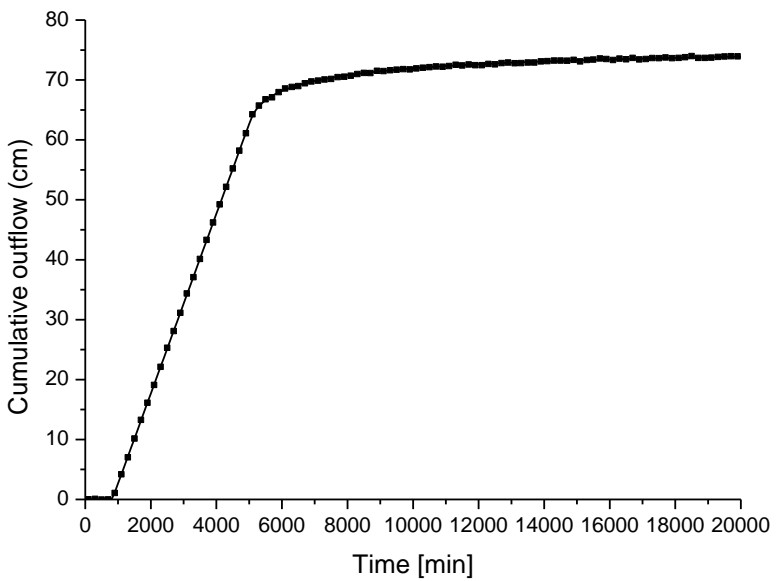

Fig. 3. Reference cumulative outflow [see Fig. 1 caption ].

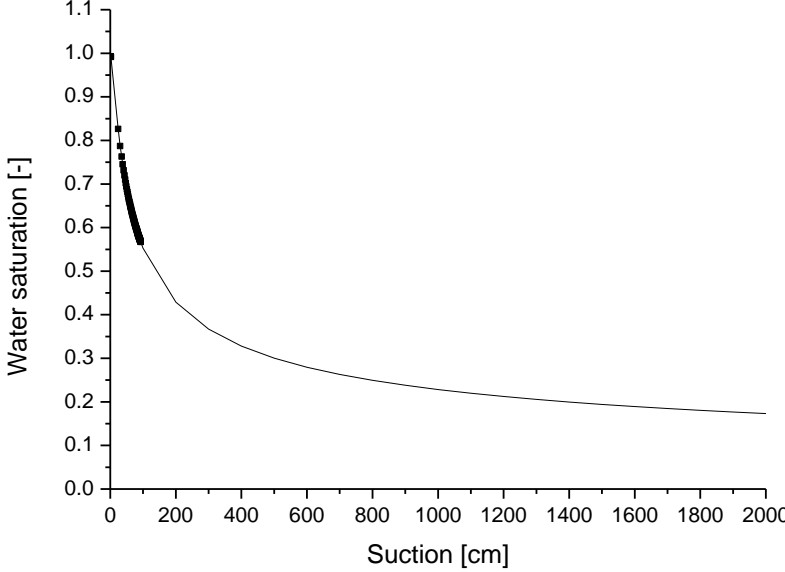

Fig. 4. Reference retention curve for the infiltration experiment [see Fig. 1 caption ].





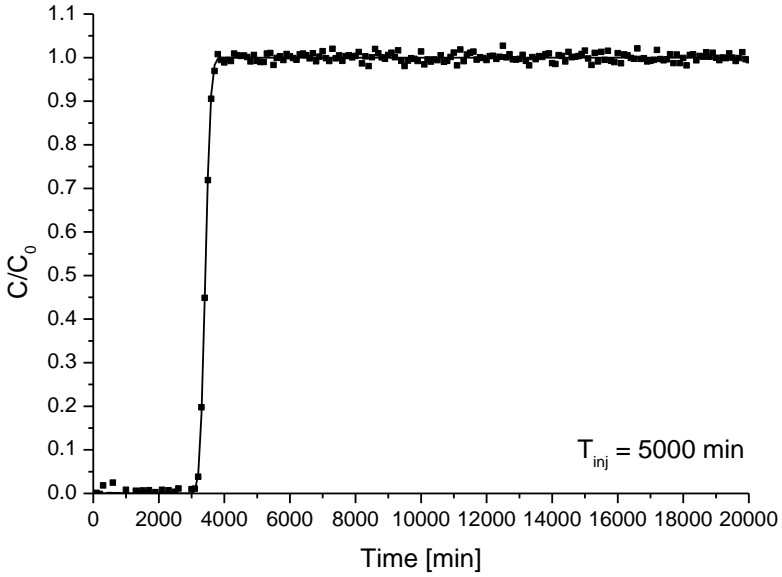

519  Fig. 5. Reference breakthrough output concentration for $T_{inj}$ = 5000. [see Fig. 1 caption ].

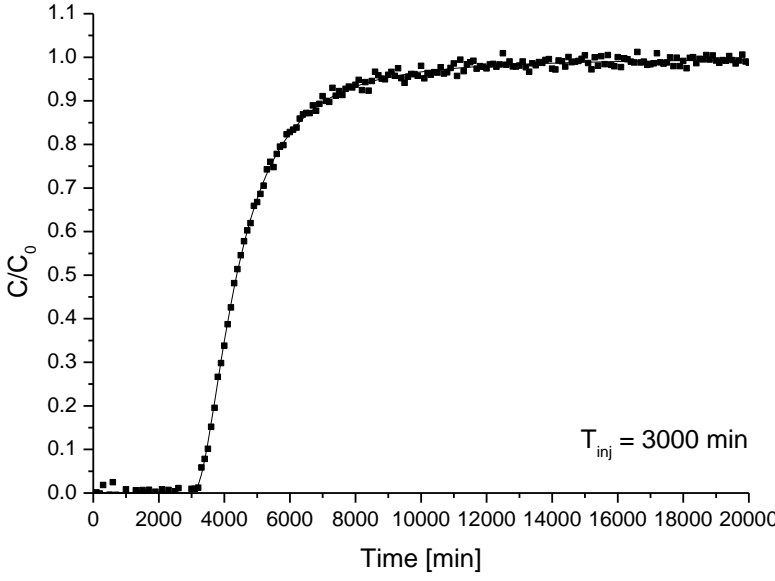

521  Fig. 6. Reference breakthrough output concentration for $T_{inj}$ = 3000 min. [see Fig. 1 caption ].





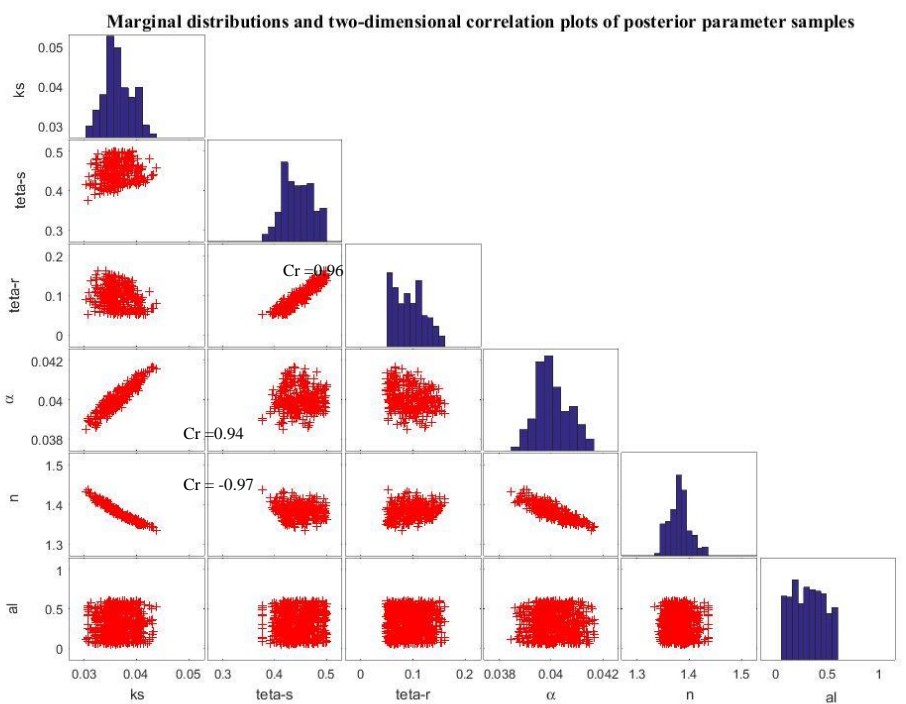

2      Fig. 7. MCMC solutions for the transport scenario 1. The diagonal plots represent the inferred posterior probability distribution of the model
3      parameters. The off-diagonal scatterplots represent the pairwise correlations in the MCMC drawing.





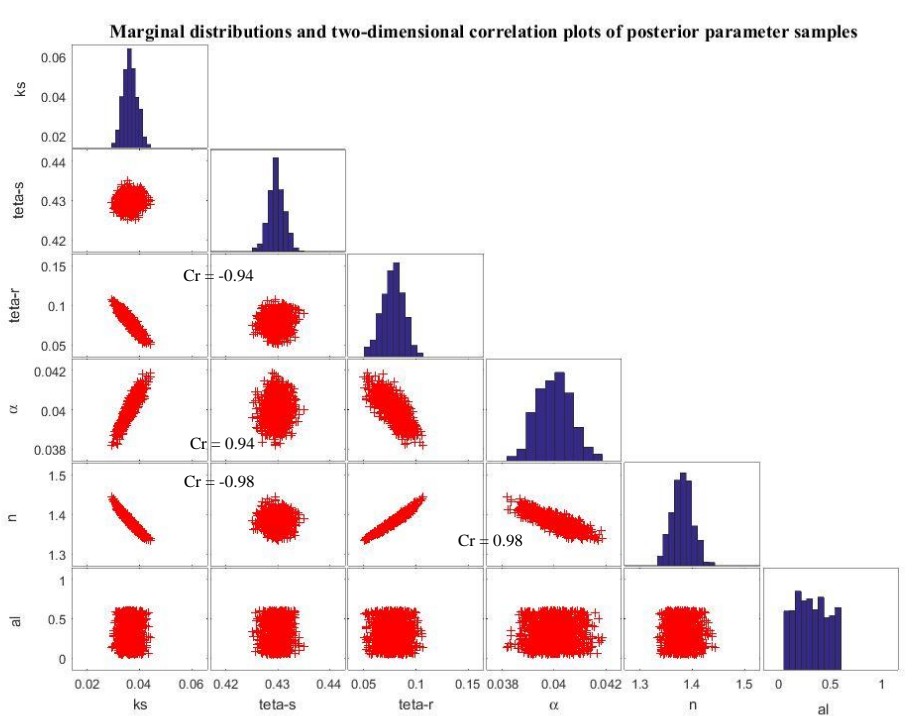





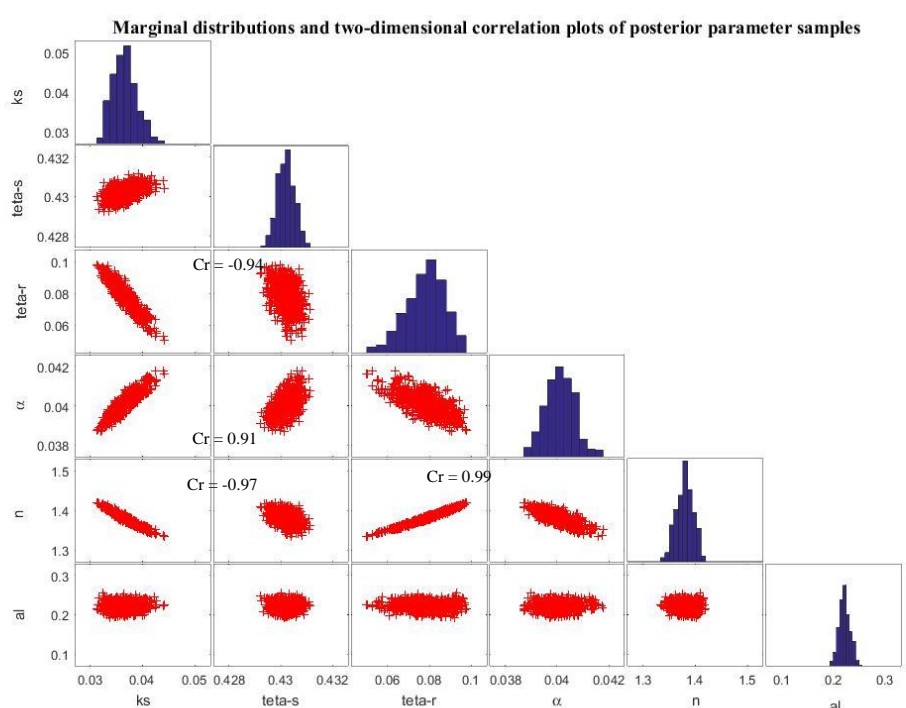

2    Fig. 9. MCMC solutions for transport scenario 3 [see Fig. 7 caption ].





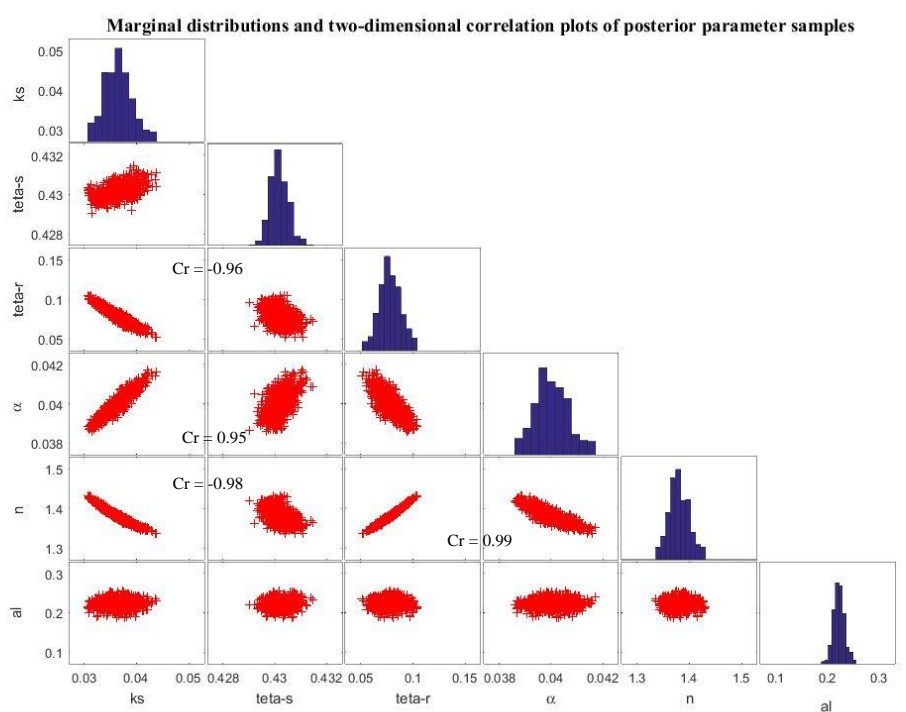

2    Fig. 10. MCMC solutions for transport scenario 4 [see Fig. 7 caption ].





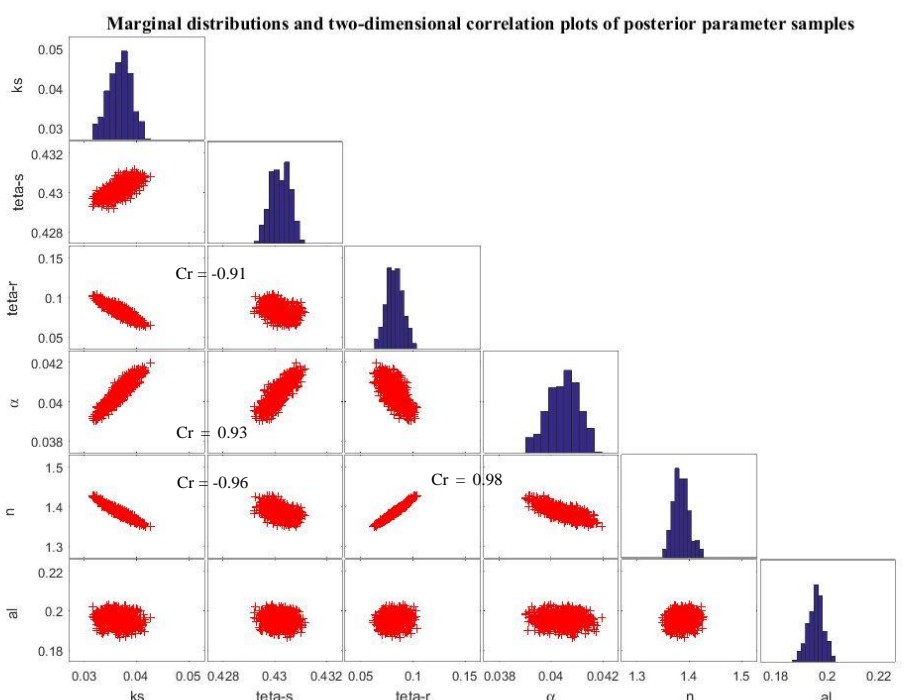

2    Fig. 11. MCMC solutions for transport scenario 5 [see Fig. 7 caption ].





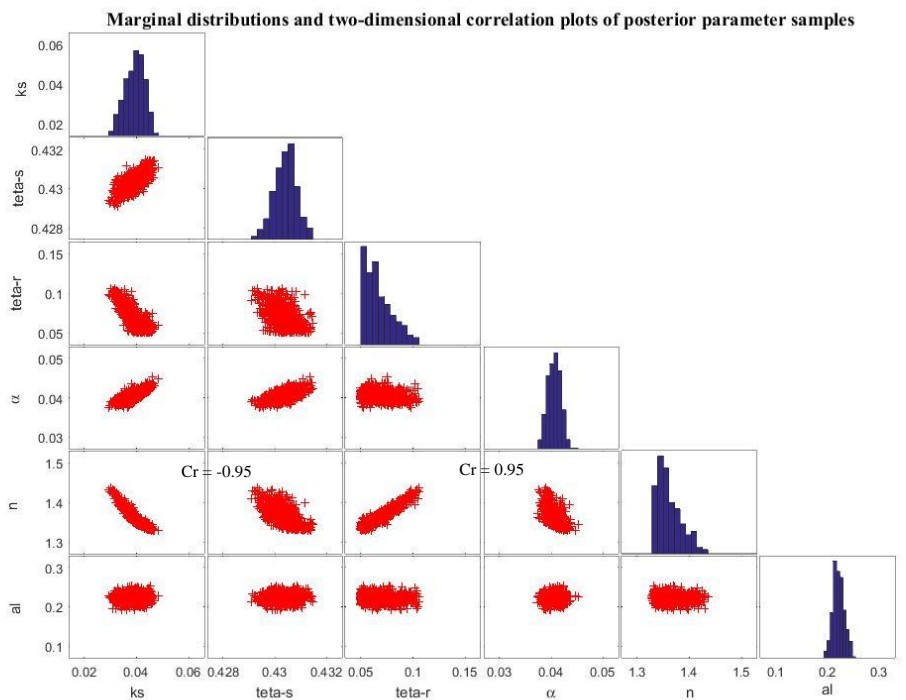

2    Fig. 12. MCMC solutions for transport scenario 6 [see Fig. 7 caption ].





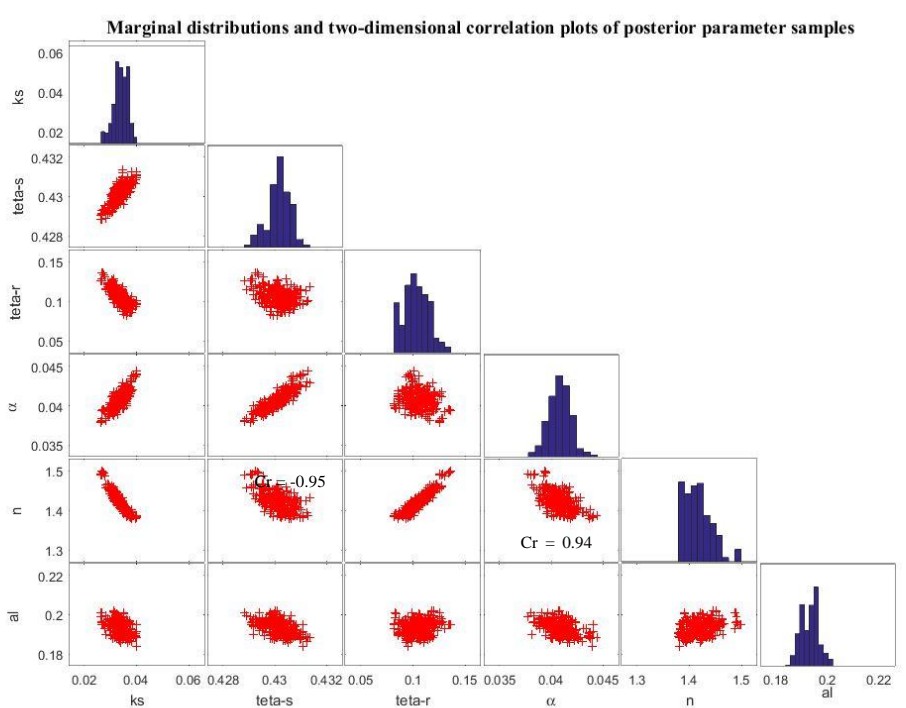

2      Fig. 13. MCMC solutions for transport scenario 7 [see Fig. 7 caption ].





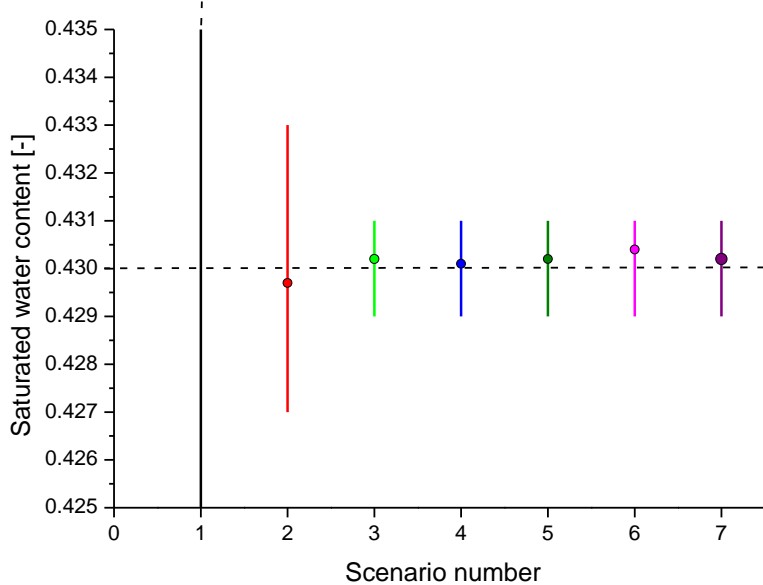

Fig. 14. Posterior mean values and 95% confidence intervals of the saturated hydraulic conductivity for the different scenarios.

Fig. 15. Posterior mean values and 95% confidence intervals of the saturated water content for the different scenarios.





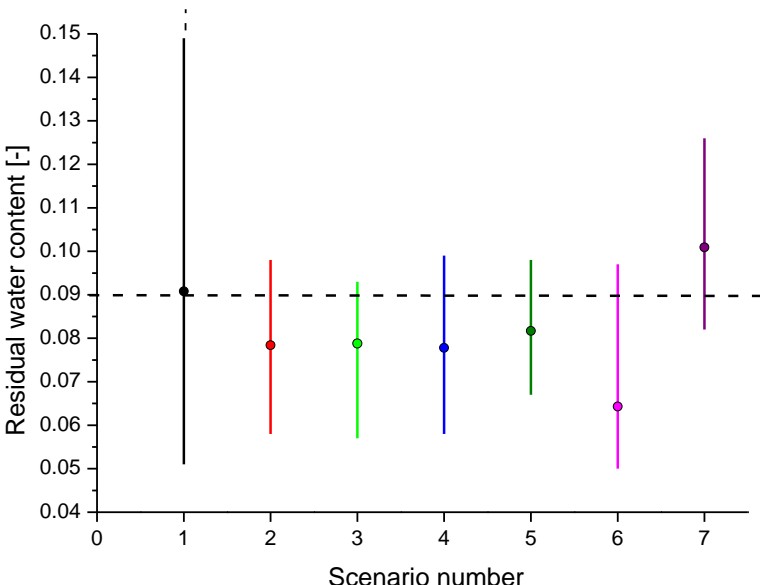

Fig. 16. Posterior mean values and 95% confidence intervals of the residual water content for the different scenarios.

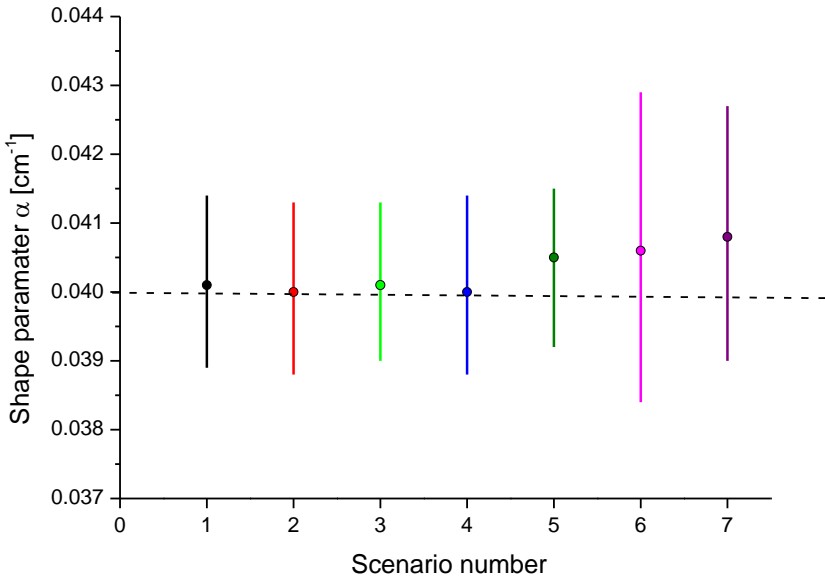

Fig. 17. Posterior mean values and 95% confidence intervals of the shape parameter □ for the different scenarios.





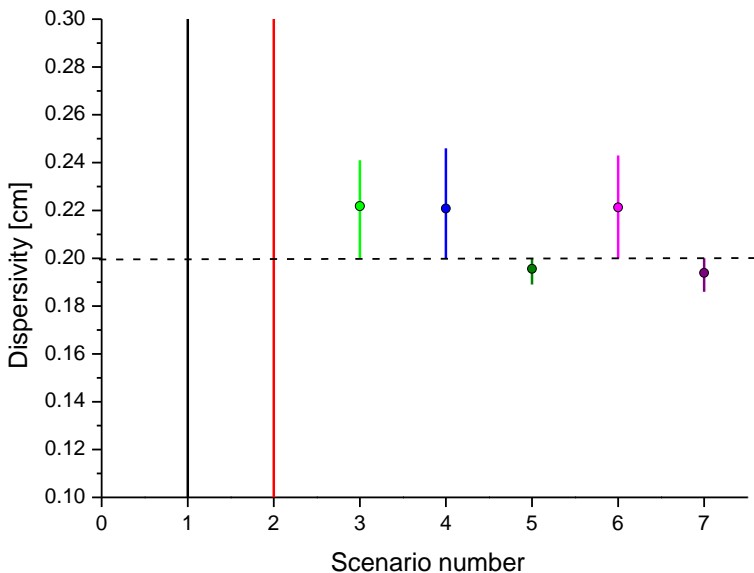

Fig. 18. Posterior mean values and 95% confidence intervals of the shape parameter n for the different scenarios.

Fig. 19. Posterior mean values and 95% confidence intervals of dispersivity for the different scenarios.