# Peer review of "Hydraulic and transport parameter assessment using column infiltration experiments"

_Hydrology and Earth System Sciences, 2016_

## Referee Comment (RC1) · Anonymous Referee #1 · 16 Aug 2016

General comment: The paper presents a study on the quality of the statistical calibration of hydraulic and transport soil properties using an infiltration experiment. In the experiment, tracer-contaminated water is injected into a laboratory column filled with a homogeneous soil in a given period. Influences of different experimental factors on the calibration results were studied.

In general, this paper deals with an interesting issue. I find some merits in the both methodology and results. As the authors describe, the soil parameters that influence water flow and contaminant transport in unsaturated zones are not generally known a priori and have to be estimated by fitting model responses to observed data. The authors realized this issue and pointed out the limitations of their work. Overall, this paper has a good potential to be published in the journal. English is also very easy to read in the manuscript. Authors have done much work and give us an exciting

theoretical and experimental study results. However, there are some issues, listed below, that need to be addressed before it is ready for publication.

Revised comment: 1. From the abstract, we want to know what you have done in your manuscript, but I can not know which parameters you have calibrated in your abstract. Please describe them in the abstract.

2. In the introduction section, please describe the development on soil parameters in more detail, and please highlight the innovation of this manuscript.

3. In the results and discussion section, please analyze in more detail.

4. In the conclusions section, please describe the further work needs to be done.

---

## Referee Comment (RC2) · Anonymous Referee #2 · 29 Aug 2016

The paper deals with an inverse modelling method determining simultaneously hydraulic and transport parameters from a packed soil column. Some of the questions posed are very useful for experimental work on flow and transport and will help future work to choose efficient experimental designs to obtain parameters. Overall the paper focusses on the methodological aspects without posing a clear hypothesis. With no clear hypothesis formulated, I would expect to have a stronger statement on the benefits of the methods employed and what we should be learning from this (not just stating that the methods used in the paper are superior over the methods other researchers have used). Even if we come up with better parameter estimation, do we have a better understanding of the physics of fluid flow in porous media? The authors should be stating what novel insights they expect from this type of numerical experiments. Furthermore, some of the findings are to be expected, for example the inclusion of both

water content or out flow along with matric potential data should always provide better parameter estimation. In fact, the use of only one of those variables makes parameter estimation non-unique. An interesting aspect of their work is the impact of the length of the injection of the solute pulse. Can the authors provide some kind of explanation why this occurs?

Considering how fractional derivatives and continues time random walk have been used to describe solute transport in unsaturated soil, will the parameter estimation method give hints on systematic model errors (which require real world experiments). Certainly one short coming of the approach - it is assumed that the model is indeed correct.

To make this paper a value contribution I suggest the following:

(i) Include a clearer summary of what has been done on inverse modelling in the context of transient water flow and solute transport. Perhaps state the methods more explicitly that were used by other researchers.

(ii) The methods sections need more precise description of numerical methods used and experimental set up. I doubt this paper is reproducible with the information provided. The language is used in such a way, that true experiments were actually done. When the authors talk about experiments they mean virtual numerical experiments. This needs to be clearly stated earlier in the paper.

(iii) The discussion section needs a thorough revision to address the above points – clearly relate your findings to the work of others on parameter estimation. Currently the discussion focusses only own findings without setting a broader context.

Further comments:

Lines 74-75: When stating column length, column diameter should also be mentioned if real world experiments were used.

Lines 83-92: The research questions are not logical derived from previous research -

they were certainly retrospectively formulated based on the findings of study.

Line 113: There is an issue with the van Genuchten - Mualem model near saturation (hydraulic conductivity will decrease before air entry point as been reached)- will this affect parameter estimation.

Lines132-139: Be precise on what was exactly implemented. The numerical scheme should be exactly described (appendix or supplemental materials are sufficient for this purpose.

СЗ

---

## Author Comment (AC1)

We thank the reviewer for their thoughtful and detailed comments that definitely helped us clarify the manuscript and avoid misinterpretations.

*General comment: The paper presents a study on the quality of the statistical calibration of hydraulic and transport soil properties using an infiltration experiment. In the experiment, tracer-contaminated water is injected into a laboratory column filled with a homogeneous soil in a given period. Influences of different experimental factors on the calibration results were studied.*

*In general, this paper deals with an interesting issue. I find some merits in the both methodology and results. As the authors describe, the soil parameters that influence water flow and contaminant transport in unsaturated zones are not generally known a priori and have to be estimated by fitting model responses to observed data. The authors realized this issue and pointed out the limitations of their work. Overall, this paper has a good potential to be published in the journal. English is also very easy to read in the manuscript. Authors have done much work and give us an exciting paper theoretical and experimental study results.*

We thank the reviewer for his/her positive overall appraisal of our work.

*However, there are some issues, listed below, that need to be addressed before it is ready for publication.*

***Revised comment:***

*1. From the abstract, we want to know what you have done in your manuscript, but I can not know which parameters you have calibrated in your abstract. Please describe them in the abstract.*

The abstract will rewritten as follows.

The quality of the statistical calibration of hydraulic and transport soil properties is studied for infiltration experiments in which, over a given period, tracer-contaminated water is injected into an hypothetical column filled with a homogeneous soil. The saturated hydraulic conductivity, the saturated and residual water contents, the Mualem-van Genuchten shape parameters and the longitudinal dispersivity are estimated in a Bayesian framework using the Markov Chain Monte Carlo (MCMC) sampler. The impact on the quality of the estimated parameters of the kind of measurement sets (water content and/or pressure inside the column, solute concentration at the outlet and cumulative outflow) and that of the injection duration of the solute is investigated by analyzing the calibrated model parameters and their confidence intervals for different scenarios. The results show that the injection period has a significant effect on the quality of the estimation, in particular, on the posterior uncertainty range of the parameters. All hydraulic and transport parameters of the investigated soil can be well estimated from the experiment using only the outlet concentration and cumulative outflow, which are measured non-intrusively. An improvement of the identifiability of the hydraulic parameters is observed when the pressure data from measurements taken inside the column are also considered in the inversion.

*2. In the introduction section, please describe the development on soil parameters in more detail, and please highlight the innovation of this manuscript.*

A significant number of references will be added and the introduction will be changed as follows:

[revised manuscript text omitted]

*3. In the results and discussion section, please analyze in more detail.*

We will provide some more explanations, especially concerning the injection duration. The explanations will be:

The improvement of the parameter estimation in this last scenario compared to the previous one can be explained by the fact that the injection of water and solute contaminant is stopped once the concentration reaches the column outlet. Hence, the injected volume (0.015x3000 = 45cm$^3$/cm$^2$) is slightly less than the pore volume (120x0.43=51 cm$^3$/cm$^2$). Thus, when the injection is stopped, the column is not fully saturated and the outlet flux strongly reduces (see the asymptotic behavior of the cumulative outflow when the injection is stopped). As a consequence, the concentration profile increases smoothly (see Fig. 6) until reaching its maximum value in contrast to the sharp front observed for $T_{inj} = 5000\,\text{min}$ in the scenario 6 (see Fig. 5). As a consequence, the breakthrough curve obtained with $T_{inj} = 3000\,\text{min}$ is more affected by the hydraulic parameters than the breakthrough curve obtained with $T_{inj} = 5000\,\text{min}$. This explains why a better estimation of the parameters is observed for the last scenario compared to the scenario 6.

*4. In the conclusions section, please describe the further work needs to be done*

Possible extensions of this work are:

These results are of course related to the models and experimental conditions we used. This work will be extended to different types of soils, water retention and/or relative permeability functions to evaluate the interest of coupling flow and transport for parameter identification. This work can also be extended to reactive solutes.

---

## Author Comment (AC2)

We thank the reviewer for his/her thoughtful and detailed comments that definitely helped us clarify the manuscript and avoid misinterpretations.

*The paper deals with an inverse modelling method determining simultaneously hydraulic and transport parameters from a packed soil column. Some of the questions posed are very useful for experimental work on flow and transport and will help future work to choose efficient experimental designs to obtain parameters. Overall the paper focusses on the methodological aspects without posing a clear hypothesis. With no clear hypothesis formulated, I would expect to have a stronger statement on the benefits of the methods employed and what we should be learning from this (not just stating that the methods used in the paper are superior over the methods other researchers have used).*

The modeling concepts were clearly stated in the introduction (L57-L61 of the submitted manuscript). The introduction has been improved and the different assumptions are described.

We did not claim that our methods are superior to methods used previously. We analyze the accuracy of some existing methods and we suggest an alternative one which avoids intrusive measurements of pressure and/or water content. We show that this new method provides quite good estimates of the parameters but, of course, not with the same accuracy than methods with intrusive measurements.

*Even if we come up with better parameter estimation, do we have a better understanding of the physics of fluid flow in porous media? The authors should be stating what novel insights they expect from this type of numerical experiments. Furthermore, some of the findings are to be expected, for example the inclusion of both water content or outflow along with matric potential data should always provide better parameter estimation. In fact, the use of only one of those variables makes parameter estimation non-unique.*

Parameter estimation through inverse modelling has a weak point: the assumption that the model is valid. Therefore, it will not provide a better understanding of the physics. It can sometimes be used to reject a model if the estimated parameters have no physical meanings.

We agree that some findings are expected. The MCMC approach allows some quantification of the uncertainties.

*An interesting aspect of their work is the impact of the length of the injection of the solute pulse. Can the authors provide some kind of explanation why this occurs?*

We will provide the following explanations in the discussion.

The improvement of the parameter estimation in this last scenario compared to the previous one can be explained by the fact that the injection of water and solute contaminant is stopped once the concentration reaches the column outlet. Hence, the injected volume ($0.015 \times 3000 = 45 \mathrm{cm}^3/\mathrm{cm}^2$) is slightly less than the pore volume ($120 \times 0.43 = 51 \ \mathrm{cm}^3/\mathrm{cm}^2$). Thus, when the injection is stopped, the

column is not fully saturated and the outlet flux strongly reduces (see the asymptotic behavior of the cumulative outflow when the injection is stopped). As a consequence, the concentration profile increases smoothly (see Fig. 6) until reaching its maximum value in contrast to the sharp front observed for $T_{inj} = 5000\,\text{min}$ in the scenario 6 (see Fig. 5). As a consequence, the breakthrough curve obtained with $T_{inj} = 3000\,\text{min}$ is more affected by the hydraulic parameters than the breakthrough curve obtained with $T_{inj} = 5000\,\text{min}$. This explains why a better estimation of the parameters is observed for the last scenario compared to the scenario 6.

*Considering how fractional derivatives and continues time random walk have been used to describe solute transport in unsaturated soil, will the parameter estimation method give hints on systematic model errors (which require real world experiments). Certainly one short coming of the approach - it is assumed that the model is indeed correct.*

The modeling concepts are assumed to be valid. See our answer to your second comment.

*To make this paper a value contribution I suggest the following:*

*(i) Include a clearer summary of what has been done on inverse modelling in the context of transient water flow and solute transport. Perhaps state the methods more explicitly that were used by other researchers.*

The introduction will be rewritten with a significant number of new references as follows:

[revised manuscript text omitted]

*(ii) The methods sections need more precise description of numerical methods used and experimental set up. I doubt this paper is reproducible with the information provided. The language is used in such a way, that true experiments were actually done. When the authors talk about experiments they mean virtual numerical experiments. This needs to be clearly stated earlier in the paper.*

The experiments are numerical experiments. This was clearly stated in the introduction (L93 of the submitted manuscript). Although we think that numerical methods for solving the flow and transport equations have to be improved, we did not addressed this issue here. The domain is 1D which does not required heavy computational equipment and standard numerical methods are accurate enough. Standard finite differences have been used for solving the equation.

All required data, initial and boundary conditions are described in the paper. The simulations can be reproduced.

*(iii) The discussion section needs a thorough revision to address the above points – clearly relate your findings to the work of others on parameter estimation. Currently the discussion focusses only own findings without setting a broader context.*

The broader context has been described in the new introduction.

***Further comments:***

*Lines 74-75: When stating column length, column diameter should also be mentioned if real world experiments were used.*

The diameter is not a relevant characteristic for our numerical examples since we use 1D simulations.

*Lines 83-92: The research questions are not logical derived from previous they were certainly retrospectively formulated based on the findings of study.*

We agree and reformulate the questions in the new introduction.

*Line 113: There is an issue with the van Genuchten - Mualem model near saturation (hydraulic conductivity will decrease before air entry point as been reached)- will this affect parameter estimation.*

We agree. However, this effect is not taken into account in this work. An extension of our work on different kind of model (Brooks and Corey, Modified Van Genuchten) is a perspective of this work.

*Lines132-139: Be precise on what was exactly implemented. The numerical scheme should be exactly described (appendix or supplemental materials are sufficient for this purpose.*

We used very standard 1D finite difference for spatial discretization. Because the method is very popular, we do not think it requires a detailed description. Details on the use of the MOL for solving RE are well described in Fahs et al. (2009). This point will be specified in the revised version.

---

## Referee Report (RR1)

Reviewer G

Hess article MS No.: hess-2016-295: **Hydraulic and transport parameter assessment using column infiltration experiments** by Younes et al. 2017  Iteration: Major Revision a manuscript after  major revision

The manuscript is deficient in the abstract, introduction, and conclusion, and lack of method session. At this late stage of the review process, the reviewer still finds it is hard to find a correct concept of what this paper about, after having finished looking through the abstract, conclusion, tables and figures, and the introduction. This reviewer gets an impression that "we" have done this but without why, how and so what. This underlined issue is in missing a conceptual model leading the discussions.

This reviewer was good with the first impressions of the article having a fine topic and of a manuscript with plenty of tables and figures, and all the mathematics. However, as he got through the reviewing process, after the first 10 minutes, he was lost in trying to find there is no description of an actual column experiment or whatever. In the text, mentioning of measurement and observations, again and again, makes this even more confusing.

In light of that the problems become so severe in structuring the text, this paper is immature. This reviewer would opt for recommending a rejection for the paper.

Detailed comments:
1. In the abstract, there is no mention of the related issues or problems with flows and transports in general. What is the approach used and what is the advantage of the approach? In the introduction section, there is no mentioning of backgrounds, the research problem, and the method.  So it is not clear why such a work is needed.

2. There is lack of discussion of method. In the beginning of the abstract, it clearly states statistically calibration of hydraulic and transport properties using an infiltration experiment with a laboratory column filled with a homogeneous soil. And then follows with "Several state variables (e.g., water content, solute concentration, pressure head) are measured during the experiment."  This review would expect to see conceptually a setup of the experimental column.

3. In the modeling session, it is one-dimensional grid for the model. How wide is the column along with the length of 1.2 meters? Any discussion how the width direction would have on tracer transport?

4. In the conclusion session, only see the list of points but no discussions on the limitations or shortcoming, and possible implications (if any).

5. There are too many figures, which some of them can be combined to be shown. Some of the figures are poor in quality with very small font sizes.

6. Section 2. A few lines below Eq. 5: $q_{inj} = 0.015$ cm/min; injection concentration $C_{inj} = 1$ g/cm$^3$. Is not this concentration too high?

The $q_{inj}$ is the Darcy velocity (Eq. 1) and is used to define the injection rate. At this rate of injection $q_{inj} = 0.015$ cm/min at the end of injection 5000 min, the water should have flown for about 75 cm. This is fine.

7. Section 3 mentions observations and measurements. Again there is no description of the "experiment"

8. Section 4.1  Reference solution and data measurements
"The pressure head at 5 cm, at the top of the column (Fig.1), increases quickly form its initial hydrostatic negative value (approximately -115 cm) and reaches a plateau (-1.75 cm) during the injection period. After the injection is finished, it progressively decreases due to the drainage caused by the gravity effect."

9. Fig. 1 has a few problems:  where is the initial -115 cm shown? No time stages marked.

Fig.1 title   The wording "reference pressure head at 5 cm from the soil surface" is confusing. Above or below?

10. Fig.3 Cumulative outflow     The outflow has significant presence at 1000 min in the plot. But based on injection rate 0.015 cm/min, the flow should have traveled for 15 cm at the time 1000 minutes. So the cumulative flow shown has a problem.

11. Fig.4 Retention curve. There is no showing for the time factor in the plot. Since it mentions reference retention curve, what is the meaning for showing suction for saturation up to 0.9? (Fig. 2 has water content about .44 at its maximum.) Is this figure showing results from one of the equation?

---

## Referee Report (RR2)

**Access review, peer review, and interactive public discussion (HESSD)**

Manuscripts submitted to HESS at first undergo a rapid access review by the editor (initial manuscript evaluation), which is not meant to be a full scientific review but to identify and sort out manuscripts with obvious deficiencies in view of the above principal evaluation criteria. Since a HESSD paper will be publicly accessible on the web, it should meet general criteria of readability. It should be well-written, well-referenced and well-structured. Figures and tables should be in good shape and referred to accordingly. In addition, the paper should contribute something new and interesting to the hydrological community.

If they are not immediately rejected, they will be published on the Hydrology and Earth System Sciences Discussions (HESSD) website, the discussion forum of HESS, where they are subject to full peer review and interactive public discussion.

In the full review and interactive discussion, the referees and other interested members of the scientific community are asked to take into account all of the following aspects:

1. Does the paper address relevant scientific questions within the scope of HESS?

   Yes

2. Does the paper present novel concepts, ideas, tools, or data?

   Yes

3. Are substantial conclusions reached?

   Yes

4. Are the scientific methods and assumptions valid and clearly outlined?

   Yes

5. Are the results sufficient to support the interpretations and conclusions?

   Yes

6. Is the description of experiments and calculations sufficiently complete and precise to allow their reproduction by fellow scientists (traceability of results)?

   Yes

7. Do the authors give proper credit to related work and clearly indicate their own new/original contribution?

   Yes

8. Does the title clearly reflect the contents of the paper?

   Yes

9. Does the abstract provide a concise and complete summary?

   Yes

10. Is the overall presentation well structured and clear?

    Yes

11. Is the language fluent and precise?

    Yes

12. Are mathematical formulae, symbols, abbreviations, and units correctly defined and used?

    Yes

13. Should any parts of the paper (text, formulae, figures, tables) be clarified, reduced, combined, or eliminated?

    No

14. Are the number and quality of references appropriate?

    Yes

15. Is the amount and quality of supplementary material appropriate?

    Yes

Review comments to "Hydraulic and transport parameter assessment using column infiltration experiments"

By A. Younes, T. A. Mara, M. Fahs, O. Grunenberger, and Ph. Ackerer

General comment:

According to response to my previous review, the authors have incorporated my suggestions into the manuscript, because the authors made all the improvements I recommended. The authors had also polished the manuscript for English spelling and terminology, and it is significantly improved from the previous version. The improved manuscript has made the methods and results understandable to the readers.

In my view, this paper deals with an interesting and highly complicated issue. The authors have completed the response in this paper. Therefore, I recommend this paper for publication in the journal.

---

## Referee Report (RR3)

HESS-2016-295 Reviewer Report

Younes et al., 2016, Hydraulic and transport parameter assessment using column infiltration experiments

General comments:

This research article introduces an inverse modeling study of the soli parameters of column infiltration experiments using MCMC sampler. General speaking, this study is novel, and provides important scientific contribution for understanding the soil parameters. I would suggest to be published in HESS after the following minor revisions and technical comments are addressed.

1) Ln 30-34: This sentence may be too long to read.
2) Ln 93-94: The benefits of the Levenberg-Marquardt algorithm should be briefly addressed here. Please refer to previous literatures.
3) Ln 117,120, 124: The number bullets should be 1), 2) and 3).
4) Ln 118-119: The authors only addressed a limited range of water content under moderately dry conditions. It is okay, but what about other conditions? Is there any specific reason that the author didn't analyze the water content in a wider range? Please provide more information.
5) Ln 151: How does the transport equation coupled with the Richard equation? Please provide more details
6) Ln 178-180: I'm confused about the reference solutions. If the reference solutions are form previous literature, please add citations. If the authors calculated the parameter values or measured from experiments, please provide more details.
7) Ln 185-187: How to determine the standard deviations? Are MCMC output and conclusion sensitive to the standard deviation values?
8) Ln 223-224: How did the authors determine the seven scenarios of measurements sets and periods of injection? Please provide more detailed information. If the design is based on previous studies, please add citation and be specified. In my understanding, the authors would like to claim that the soil parameters can be better estimated by C instead of $\theta$, and non-intrusive measurements are good enough for parameter estimations. I hope to see a clear and specific explanation of the reasons for each scenario, highlighted before discussing the result of each scenario. For example, scenarios 2 and 4 compared the effect of C and $\theta$ to parameter estimation.
9) I would suggest to re-organize Fig. 1-7 as one panel figure, since they all present the match of simulation and observed data, and the individual plot may stand too much space in publication.
10) Ln 286-291: The authors explain that $(\theta_r, \theta_s) = 0.96$ is only observed and cannot be identified accurately in scenario 1, when water contents are not evaluated. I would suggest to highlight it in Table 3 as well in case if readers miss the text.
11) How did you make conclusion 4) (the estimation of the dispersivity is sensitive to the injection duration)? I didn't see discussion about this point before conclusion. Please specified the scenarios you based to draw this conclusion.

12) The readers can understand the conclusion more straightforward, if the sources of each conclusion are highlighted. For example, conclusion 5) is from a comparison between scenarios 2 and 4.

13) I agree with the comment ii) from reviewer 2. The authors should clearly state that the experiments are numerical experiment in the paper to avoid misleading. The revised manuscript still does not have a clear statement and description about the experiment and the observed data used in this study. This must be done before the paper can be published.

---

## Author Response (AR2)

Review comments to "Hydraulic and transport parameter assessment using column infiltration experiments"
By A. Younes, T. A. Mara, M. Fahs, O. Grunenberger, and Ph. Ackerer

**General comment:**
*According to response to my previous review, the authors have incorporated my suggestions into the manuscript, because the authors made all the improvements I recommended. The authors had also polished the manuscript for English spelling and terminology, and it is significantly improved from the previous version. The improved manuscript has made the methods and results understandable to the readers.*

*In my view, this paper deals with an interesting and highly complicated issue. The authors have completed the response in this paper. Therefore, I recommend this paper for publication in the journal.*

We thank the referee for his review whose constructive comments helped in improving the manuscript. We are of course pleased that she/he considers now that the manuscript presents a novel study, and that it is well thought out and well structured.

**Anonymous Referee #3**

Review comments to "Hydraulic and transport parameter assessment using column infiltration experiments"
By A. Younes, T. A. Mara, M. Fahs, O. Grunenberger, and Ph. Ackerer

Line numbers in red referred to the revised new manuscript, in blue to the reviewed manuscript.

*The manuscript is deficient in the abstract, introduction, and conclusion, and lack of method session.*
> The initial and the revised manuscript have been reviewed by 4 other referees and the editor. No one complains about the abstract, introduction, …

*At this late stage of the review process, the reviewer still finds it is hard to find a correct concept of what this paper about, after having finished looking through the abstract, conclusion, tables and figures, and the introduction. This reviewer gets an impression that "we" have done this but without why, how and so what. This underlined issue is in missing a conceptual model leading the discussions.*
> The objectives are described (see lines **L82-L84** of the submitted manuscript).
> The methodology is addressed **L93-L106**.
> Three questions are addressed **L117-L124.**

*This reviewer was good with the first impressions of the article having a fine topic and of a manuscript with plenty of tables and figures, and all the mathematics. However, as he got through the reviewing process, after the first 10 minutes, he was lost in trying to find there is no description of an actual column experiment or whatever. In the text, mentioning of measurement and observations, again and again, makes this even more confusing.*
> It is clearly stated from the beginning that the experiments are not real **(L107)** but we will rephrase this sentence. It is also mentioned in the abstract (**L27**), we will also re-write it.
> We also mentioned that the scenarios are synthetic (i.e. not real) (**L85, L125, L230, L373**).
> These measured values are generated as explained **L184-L187**.
> See also the **title of § 4.1…**
> We changed **L178 to L187** to provide more details, see L182-1998.

*In light of that the problems become so severe in structuring the text, this paper is immature. This reviewer would opt for recommending a rejection for the paper.*
> No comment.

**Detailed comments:**
*1. In the abstract, there is no mention of the related issues or problems with flows and transports in general. What is the approach used and what is the advantage of the approach? In the introduction section, there is no mentioning of backgrounds, the research problem, and the method. So it is not clear why such a work is needed.*
> The paper deals with statistical estimations of hydraulic and transport soil properties (**L25**). The methodology is explained **L99-L106** and we address three questions (**L117-L124**) which describe the research problem.

*2. There is lack of discussion of method. In the beginning of the abstract, it clearly states statistically calibration of hydraulic and transport properties using an infiltration experiment with a laboratory column filled with a homogeneous soil. And then follows with "Several state variables (e.g., water content, solute concentration, pressure head) are measured during the experiment." This review would*

*expect to see conceptually a setup of the experimental column.*
Again, it is not a real experiment (**L85, L125, L184-L187, L230, L373, title of § 4.1**).
The setup and initial conditions are explained **L107-L114**.
We changed **L178 to L187** to provide more details, see L182-L198.

*3. In the modeling session, it is one-dimensional grid for the model. How wide is the column along with the length of 1.2 meters? Any discussion how the width direction would have on tracer transport?*
The column width is not relevant. This is well known for one dimensional problem.

*4. In the conclusion session, only see the list of points but no discussions on the limitations or shortcoming, and possible implications (if any).*
Limitations are given at the end of the conclusion.
We provide also practical applications for designing experimental set-ups (**L397-L403**).

*5. There are too many figures, which some of them can be combined to be shown. Some of the figures are poor in quality with very small font sizes.*
We provide numerous results and figures help in understanding. Figures 7 to 13 have been improved. Figures 1 to 6 have been merged in one figure.

*6. Section 2. A few lines below Eq. 5: $q_{inj} = 0.015$ cm/min; injection concentration $C_{inj} = 1$ g/cm3. Is not this concentration too high?*
It is a numerical solution. For real conditions, it would be probably too high and density driven flow may appear. Here, we are in the context of a tracer experiments based on equations (1), (2) and (3).

*The $q_{inj}$ is the Darcy velocity (Eq. 1) and is used to define the injection rate. At this rate of injection $q_{inj}$ = 0.015 cm/min at the end of injection 5000 min, the water should have flown for about 75 cm. This is fine.*
Not correct. The reviewer does not make the difference between Darcy's velocity and mean pore water velocity. Darcy's velocity has to be divided by the water content …

*7. Section 3 mentions observations and measurements. Again there is no description of the "experiment"*
Again, it is a numerical experiment. See answer to comment 2.

*8. Section 4.1   Reference solution and data measurements*
*"The pressure head at 5 cm, at the top of the column (Fig.1), increases quickly form its initial hydrostatic negative value (approximately -115 cm) and reaches a plateau (-1.75 cm) during the injection period. After the injection is finished, it progressively decreases due to the drainage caused by the gravity effect."*
*Fig. 1 has a few problems:   where is the initial -115 cm shown? No time stages marked.*
There is no problem in figure 1. The initial pressure at 5 cm is -115 cm. This cannot be seen in the figure because pressure near the surface increases very quickly (less than 100 min) to -1.75 cm due to the high injection rate. It is better described in the revised version, L250-L252.

*9. Fig.1 title    The wording "reference pressure head at 5 cm from the soil surface" is confusing. Above or below?*
It is below (above, there is no porous material). We changed the text accordingly, L188-L190.

*10. Fig.3 Cumulative outflow       The outflow has significant presence at 1000 min in the plot. But based on injection rate 0.015 cm/min, the flow should have traveled for 15 cm at the time 1000 minutes. So the cumulative flow shown has a problem.*

Not correct for two reasons: (*i*) Again, the reviewer does not make the difference between Darcy's velocity from the mean pore water velocity and (*ii*) the water exiting the column at 1000 min is not the injected water from the surface but the initial water retained in the unsaturated column. Indeed, the infiltrated water pushes the initial water in the column which explains the outflow at 1000 min. This is very well known for drainage experiments.

*11. Fig.4 Retention curve. There is no showing for the time factor in the plot. Since it mentions reference retention curve, what is the meaning for showing suction for saturation up to 0.9? (Fig. 2 has water content about .44 at its maximum.) Is this figure showing results from one of the equation?*

There is no time factor. We plot equation (2) to highlight the non-linear properties of the retention curve and to show that we do not have very dry conditions (water saturation above 0.55) as stated in the manuscript (**L241-244**).

**Anonymous Referee #4**

Review comments to "Hydraulic and transport parameter assessment using column infiltration experiments"
By A. Younes, T. A. Mara, M. Fahs, O. Grunenberger, and Ph. Ackerer

We thank the reviewer for his/her thoughtful and detailed comments that helped us to improve the manuscript. Line numbers in red referred to the revised new manuscript, in blue to the reviewed manuscript.

*Ln 30-34: This sentence may be too long to read.*
    The sentence has been modified (L31-L34).

*2) Ln 93-94: The benefits of the Levenberg-Marquardt algorithm should be briefly addressed here. Please refer to previous literatures.*
    We have referred to previous literature in this field, Gallaher and Doherty, 2007 (L95-L96).

*3) Ln 117,120, 124: The number bullets should be 1), 2) and 3).*
    It was numbered in our version of the submitted manuscript. We checked the revised version.

*4) Ln 118-119: The authors only addressed a limited range of water content under moderately dry conditions. It is okay, but what about other conditions? Is there any specific reason that the author didn't analyze the water content in a wider range? Please provide more information.*
    Moderately dry conditions are investigated because the bottom of the soil column is exposed to the atmosphere (gravity drainage). Note that a wider range of water content should improve identifiability of the parameters as suggested by Kool and Parker (1988).
    This is addressed in the revised version, L121-L122.

*5) Ln 151: How does the transport equation coupled with the Richard equation? Please provide more details*
    The transport equation (3) is coupled with the flow equation (1) by the water content and the Darcy velocity used in (3).
    This is addressed in the revised version, L158-L159.

*6) Ln 178-180: I'm confused about the reference solutions. If the reference solutions are form previous literature, please add citations. If the authors calculated the parameter values or measured from experiments, please provide more details.*
    By reference solution, we mean the solution of the inverse problem i.e. the measured/observed values. These measured values are generated as explained L184-L187. We modified L178 to L183 to better explain the way of generating observations for these numerical experiments, L182-198.

*7) Ln 185-187: How to determine the standard deviations? Are MCMC output and conclusion sensitive to the standard deviation values?*
    The measurements are performed with the following errors: $\pm 2\,cm$ for the pressure head, $\pm 0.04$ for the water content, $\pm 0.2\,cm$ for the cumulative outflow and $\pm 0.02\,g/cm^3$ for the outlet concentration (see L184 to L187). For Gaussian distributions, the 95% Confidence Interval characterizing measurement errors is $\pm 1.96\sigma$. Hence, the standard deviations are:

$\sigma_h = 1cm$ for the pressure head, $\sigma_\theta = 0.02$ for the water content, $\sigma_Q = 0.1$ cm for the cumulative outflow and $\sigma_C = 0.01$ g/cm$^3$ for the exit concentration.

*8) Ln 223-224: How did the authors determine the seven scenarios of measurements sets and periods of injection? Please provide more detailed information. If the design is based on previous studies, please add citation and be specified. In my understanding, the authors would like to claim that the soil parameters can be better estimated by C instead of θ, and non-intrusive measurements are good enough for parameter estimations. I hope to see a clear and specific explanation of the reasons for each scenario, highlighted before discussing the result of each scenario. For example, scenarios 2 and 4 compared the effect of C and θ to parameter estimation.*

We agree and added the following detailed information.
In the first scenario, only measured pressure heads and cumulative outflow are used for the calibration. The scenarios 2 to 5 investigate the benefit of adding measured water contents and/or solute outlet concentrations to pressure heads and outflow. The last scenarios (6, 7) investigate the use of only measured cumulative outflow and concentration breakthrough at the column outflow because these measurements do not require intrusive techniques. Scenarios 5 to 7 investigate as well the effects of solute injection duration on the identifiability of the parameters. L233-L240.

*9) I would suggest to re-organize Fig. 1-7 as one panel figure, since they all present the match of simulation and observed data, and the individual plot may stand too much space in publication.*

Done in the revised version.

*10) Ln 286-291: The authors explain that (θr, θs) = 0.96 is only observed and cannot be identified accurately in scenario 1, when water contents are not evaluated. I would suggest to highlight it in Table 3 as well in case if readers miss the text.*

The correlation is indicated in Table 3.

*11) How did you make conclusion 4) (the estimation of the dispersivity is sensitive to the injection duration)? I didn't see discussion about this point before conclusion. Please specified the scenarios you based to draw this conclusion.*

We agree. Dispersivity is very well estimated for scenario 5 and 7, and well estimated for scenario 3, 4 and 6. We developed conclusion, 4), L409-L412.

12) The readers can understand the conclusion more straightforward, if the sources of each conclusion are highlighted. For example, conclusion 5) is from a comparison between scenarios 2 and 4.

We referenced the scenarios in the conclusion of the revised manuscript.

13) I agree with the comment ii) from reviewer 2. The authors should clearly state that the experiments are numerical experiment in the paper to avoid misleading. The revised manuscript still does not have a clear statement and description about the experiment and the observed data used in this study. This must be done before the paper can be published.

It is clearly stated from the beginning that the experiments are not real (L107). It is also mentioned in the abstract (L27.
We also mentioned that the scenarios are synthetic (i.e. not real) (L85, L125, L230, L373). These measured values are generated as explained L184-L187.
See also the title of § 4.1…
We changed L178 to L187, see L182-198.

[revised manuscript text omitted]

Fig. 1813. Posterior mean values and 95% confidence intervals of the shape parameter n for the different scenarios.

[Figure]

Fig. 1914. Posterior mean values and 95% confidence intervals of dispersivity for the different scenarios.